# WiFi Indoor Location Based on Area Segmentation

**DOI:** 10.3390/s22207920

**Published:** 2022-10-18

**Authors:** Yanchun Wang, Xin Gao, Xuefeng Dai, Ying Xia, Bingnan Hou

**Affiliations:** 1School of Communication and Electronic Engineering, Qiqihar University, Qiqihar 161000, China; 2School of Computer and Control Engineering, Qiqihar University, Qiqihar 161000, China

**Keywords:** indoor positioning, area segmentation, deep neural networks, fingerprint database

## Abstract

Indoor positioning is the basic requirement of future positioning services, and high-precision, low-cost indoor positioning algorithms are the key technology to achieve this goal. Different from outdoor maps, indoor data has the characteristic of uneven distribution and close correlation. In areas with low data density, in order to achieve a high-precision positioning effect, the positioning time will be correspondingly longer, but this is not necessary. The instability of WiFi leads to the introduction of noise when collecting data, which reduces the overall performance of the positioning system, so denoising is very necessary. For the above problems, a positioning system using the DBSCAN algorithm to segment regions and realize regionalized positioning is proposed. DBSCAN algorithm not only divides the dataset into core points and edge points, but also divides part of the data into noise points to achieve the effect of denoising. In the core part, the dimensionality of the data is reduced by using stacking auto-encoders (SAE), and the localization task is accomplished by using a deep neural network (DNN) with an adaptive learning rate. At the edge points, the random forest (RF) algorithm is used to complete the localization task. Finally, the proposed architecture is verified on the UJIIndoorLoc dataset. The experimental results show that our positioning accuracy does not exceed 1.5 m with a probability of less than 87.2% at the edge point, and the time is only 32 ms; the positioning accuracy does not exceed 1.5 m with a probability of less than 98.8% at the core point. Compared with indoor positioning algorithms such as multi-layer perceptron and K Nearest Neighbors (KNN), good results have been achieved.

## 1. Introduction

In the information times, people’s demand for positioning is getting higher and higher. For the outdoors, many outdoor positioning devices and technologies can already meet people’s daily needs, but indoor positioning services have been developing [1]. For indoor positioning, the current technology with the highest accuracy is Simultaneous Localization and Mapping (SLAM) technology [2,3]. Although it can update the map in real-time and has high accuracy, but this technology is expensive and complicated to implement [4]. Compared with SLAM, the large-scale popularity of WiFi has promoted the rapid entry of WiFi-based positioning systems into people’s field of vision, and because of its advantages of convenient use, low cost, and relatively high accuracy, it has quickly replaced ultra-wide band (UWB) [5,6], Zigbee [7,8] and Bluetooth [9,10] have become the most widely used technologies in indoor positioning [11,12,13]. With the full popularization of 5G technology [14], WiFi-based positioning technology will gradually become the mainstream positioning technology.

At present, indoor positioning technology has been applied in many fields. For example, in the building domain, many interesting localization techniques have been proposed in building energy management [15], occupancy detection [16], and intelligent building control [17]. In the medical field, indoor positioning technology can provide basic technical support for hospitals to realize intelligent perception and processing of medical objects (such as doctors, nurses, patients, equipment, drugs, etc.) [18,19,20].

WiFi-based positioning technology is mainly divided into two categories: triangle positioning method and fingerprint positioning method [21,22]. Triangulation is also the basic operating principle of GPS [23]. It uses the distance between the measured point and the reference point for positioning. The accuracy largely depends on the accurate model of wireless signal transmission loss. However, due to the instability of WiFi [24], the movement of objects and people and even temperature changes will interfere with it, there is no particularly accurate model for a long time, so the accuracy of this method is relatively general and cannot satisfy people’s daily needs. Although some scholars now propose to combine this method with other traditional methods to improve the accuracy, this will undoubtedly increase the cost [25,26]. Fingerprint-based positioning technology collects Received Signal Strength Indicator (RSSI) values from different Access Point (AP) points in the offline phase to build a database (also known as a fingerprint database) [27,28,29]. When a user initiates a positioning request, the RSSI value of the point is matched with the data in the database to complete the positioning task. Therefore, the technology is suitable for large buildings and can meet the daily needs of the public [30,31].

A good positioning algorithm can have a significant impact on a positioning system [32,33,34], but from the current perspective, there is little scope for improving positioning accuracy through algorithms. However, it is often overlooked that the quality of the database also has an important impact on the positioning accuracy, as many external factors, such as human walking and object movement, can affect the RSSI values collected when data is being collected. Therefore, it is necessary to filter the data to remove outliers and create a high-quality database. In addition, if only one positioning algorithm is used for the positioning task, the positioning time is often too long and time is wasted in uncommon or relatively open areas, where there is no requirement for positioning accuracy. Therefore, it is necessary to identify these areas in the dataset and use a different localization algorithm. Finally, the learning rate is always one of the most difficult parameters to adjust in neural networks. We incorporated an adaptive learning rate algorithm into the DNN, which helped to obtain a better localization model in a short time.

Aiming at the above problems, a new positioning system is proposed. Compared with the previous systems, the main contributions of this paper are as follows:(1)Since people usually place WiFi in a specific place to obtain a stronger and denser WiFi signal in commonly used places, this habit is used to divide the dataset into core points and edge points according to the signal density of WiFi. Different positioning algorithms are used for different regions, which can achieve higher positioning accuracy in commonly used places and save positioning time in less commonly used places.(2)The adaptive learning rate algorithm is integrated into the DNN network to solve the problem that the learning rate parameter is difficult to adjust. At the same time, the stacked autoencoder (SAE) is used to reduce the dimension of the data to solve the problem that the dimension of the dataset is too high. Using a positioning algorithm that incorporates an adaptive learning rate algorithm and DNN network with SAE as the core point, the positioning accuracy does not exceed 1.5 m with a probability of less than 98.8%.(3)Since we use a high-precision positioning algorithm at the core point, which sacrifices the positioning time, we use the random forest algorithm as the positioning algorithm at the edge point. Although the positioning accuracy does not exceed 1.5 m with a probability of less than 87.2%, the positioning time is only 32 ms. If the user is at the edge point, the location information can be obtained in a short time.

The rest of this paper is organized as follows: the second part introduces the basic framework and related literature of WiFi positioning; the third part introduces the used dataset and system architecture; the fourth part introduces the experimental procedure and results; finally, the fifth part describes the conclusions, limitations and future work.

## 2. Related Work

In this section, the basic framework of WiFi positioning and literature related to this paper are introduced.

### 2.1. WiFi Positioning System Framework

In the previous fingerprint-based WiFi indoor positioning system, it is mainly divided into two stages: the offline stage and the online stage [35]. As shown in Figure 1 below, the offline phase mainly collects the signal strength of each point and collects the geographic location to establish a fingerprint database. Through data training, a positioning model used in the online phase is obtained, which reflects the mapping relationship between signal strength and physical location. In the online stage, the instantaneous RSSI signal obtained by the user’s online positioning request is input into the positioning model to obtain the physical position, thereby realizing indoor positioning [36,37,38].

### 2.2. Related Literature

At present, the positioning technology based on WiFi has become a popular positioning technology [39,40]. Many scholars have also realized that relying on algorithms alone to improve positioning accuracy has little effect. At present, how to build an efficient and accurate fingerprint database and how to improve the accuracy on the basis of the database have become the main bottlenecks of this technology, so there is more and more research on improving the quality of data and new positioning systems.

Tao Y et al. proposed a positioning system that can automatically build a radio map and locate online step by step. The positioning system can capture the WiFi data packets transmitted in the WiFi traffic, obtain the MAC address, frequency and RSSI of any WiFi access point, and use the Gaussian process regression model based on the firework algorithm to simulate the RSSI distribution of the indoor environment to estimate the location of the AP [41]. To address the issue of inconsistent WiFi signal observations, Du X et al. studied the signal patterns between WiFi signals and coexisting access points under modern enterprise WiFi infrastructure and the correlation of signals with indoor path maps. Firstly, the concept of signal pattern (SSP) is used and processed to generate beacon ap with high positioning reliability. During the positioning process, the estimated position is brought into a limited area through signal coverage constraint (SCC), thereby improving positioning accuracy [42]. Coincidentally, Zhang W et al. proposed an AP selection algorithm based on multi-objective optimization, which improved the accuracy of the positioning system by constructing an efficient and accurate fingerprint database [43].

Scholars have not only focused on building high-quality databases but also proposed many new indoor positioning systems. While developing a new sparse Bayesian learning algorithm to model radio power maps in indoor spaces, Ko C H et al. also propose a two-stage localization method: the localization task is divided into coarse localization and fine-grained localization. The average positioning error of this method is 1.98 m, which is 22% higher than the traditional RSSI-based positioning method [44]. While using hierarchical positioning, Tao Y et al. proposed an accurate indoor positioning system (AIPS) based on fingerprint adaptation for the problem that WiFi is easily affected by the indoor dynamic environment. In the fingerprint database update process, the K-means algorithm was used. Divide the data into two categories based on the number of loops, find out which data changes with the dynamic environment, then update it through Gaussian process regression [45]. Belmonte-Fernández Ó et al. proposed to directly calculate the WiFi wireless map through the radiosity signal propagation model to replace the manual data collection stage [46].

Neural networks have been widely used in indoor positioning systems and show good performance. Li L et al. proposed SmartLoc, a smart wireless indoor positioning framework, to enhance indoor positioning. In the offline phase, multiple machine learning models are trained using the offline database, and probabilistic alignment is applied to ensure the prediction probability of each model at the same confidence level. In the online phase, labels with probabilities greater than a certain threshold are extracted from each model to construct the size of candidate labels (SCL) determined by using the Dynamic size Determination (DSD) algorithm. Finally, they also propose a probabilistic model to estimate the user’s location by simultaneously assessing the trustworthiness of the tags. Experimental results in a real changing environment verify the superiority of SmartLoc, outperforming the best among comparative methods by 10.8% in 75th percentile accuracy [47]. Qin F et al. proposed a localization system based on convolutional denoising autoencoder (CDAE) and convolutional neural network (CNN). Compared with the traditional RSSI-based localization method, the localization accuracy of the system reached 1.05 m [48]. Using channel state information (CSI) as input, Wang X et al. used a deep convolutional neural network for product localization. After extensive experiments in two representative indoor environments, it was verified that the method has good performance [49].

Although many wireless positioning technologies have been adopted by other researches due to their low cost or high accuracy, such as RFID [50,51], BLE [52,53] UWB [54,55] and other technologies, BLE technology is considered to replace WiFi positioning technology in the future, but at present WiFi technology still has an irreplaceable position. RFID technology is a wireless technology that retrieves data from nearby transponders. Although this technology has been widely used in shopping malls, warehouses, and factories because of its energy-saving and durable characteristics, the technology has its own dedicated infrastructure (RFID reading Card reader and tag), is not supported by any mobile device, so high cost is a key factor restricting the development of RFID indoor positioning technology. For BLE technology, although it is a more suitable technology for indoor positioning than WiFi in terms of energy consumption and scanning rate, the accuracy of this technology is proportional to the number of beacons, which means that to exceed the accuracy of WiFi indoor positioning technology requires more costs, and the BLE signal is more susceptible to channel gain and rapid fading. The BLE measurement value will shake violently over time. Additionally, as mobile beacons are battery powered, ensuring uninterrupted service remains a major challenge. UWB technology is a wireless technology with high transmission rate, low transmit power, strong penetrating ability and is based on an extremely narrow pulse. The high delay resolution of this technology determines that it has multi-path recovery capabilities, but it is widely used in the location of soldiers on the battlefield, robot motion tracking, etc., and is rarely used in indoor positioning.

## 3. Proposed System Structure

In this section, the algorithm used to process the data is introduced after a brief description of the data set used, then the principles of the RF algorithm and the DNN algorithm are introduced respectively, including the derivation of the adaptive learning rate, and finally the whole positioning is introduced process.

### 3.1. Dataset Introduction

In the simulation, the effectiveness of the proposed system is verified using the UJIIndoorLoc dataset, which is located in a building of nearly 110,000 square meters in Jaume I University, Spain, with 25 different Android devices used by 18 different users Complete the collection [56].

The entire database contains 21,049 records, and each record contains the following 529 elements:
001~520 RSSI Levels521~523 Real world coordinates of the sample points
524 Building ID525 Space ID526 Relative position with respect to Space ID527 User ID528 Phone ID529 Timestamp

(1)RSSI Levels: The more important information in the WiFi information is the detected RSSI value. 98% of the data in the database belong to the RSSI value, of which −100 dbm is equivalent to a very weak signal, which can be considered as the point where no signal is detected, while 0 dbm means that a very good signal is detected.(2)Real world coordinates: Vectors 521 to 523 record longitude coordinates, latitude coordinates, and the floor of the building.(3)Space identifiers: Building ID, Vector 524 is the integer value (from 0 to 2) corresponding to the building from which the RSSI value is obtained. Space ID, Vector 525 is used to identify in a specific space (office, laboratory, etc.). Relative position with respect to Space ID, Vector 526 is used to indicate whether the location where the RSSI value is obtained is in the interior space of the corridor.(4)User ID: Containing an integer value from 1 to 18, the vector 527 is used to represent the 18 different users who collected RSSI values.(5)Phone ID: Vector 528 contains different integer values to represent different Android devices used to collect RSSI values.(6)Timestamp: The vector 529 is a timestamp, which is used to represent the time when the RSSI value was collected (in Unis time format).

In the process of actual use, the vectors such as Space ID and Relative position with respect to Space ID have a value of 0, which means that it was not recorded at that time rather than non-existent.

### 3.2. Data Processing Algorithms

Since the representation of different RSSI values will affect the positioning accuracy, the data needs to be normalized and preprocessed first. In the dataset used, the value range of RSSI data is −100 dBm~0 dBm, and the RSSI value at the AP, where no data is detected, is marked as 100 dBm. For this, we convert these values to the (0, 1) range using Equation (1) below.
(1)$$RSSI\prime =\left\{ \begin{array}{l} \frac{RSSI_{i}-min}{max-min}\;RSSI\;exists\\ 0\;\;\;\;otherwise \end{array}\right.$$
where *RSSI_i_* is the intensity value provided by the *i*th WAP, *min* is the smallest RSSI value in the dataset, and *max* is the largest RSSI value in the dataset.

Secondly, in the offline phase, for a larger area, a large number of signal strength values need to be collected to build a database to improve the positioning accuracy. Since many factors such as temperature, humidity, and people’s movement will affect WiFi, a large amount of noise will inevitably be introduced during the collection process. Therefore, a denoising algorithm needs to be used to improve the quality of the database.

The clustering algorithm can divide the data set into different clusters and can achieve the effect of denoising while finding the data with strong correlation. The commonly used algorithms are K-Means, BIRCH, CURE, DBSCAN, etc. [57,58]. The BIRCH algorithm is a balanced iterative reduction clustering algorithm, the CURE algorithm is a clustering algorithm using representative points, the K-means algorithm is a clustering algorithm for iterative solutions, and the DBSCAN algorithm is a density-based clustering algorithm. Since the fingerprint database must be irregular, however, the BIRCH algorithm and the K-means algorithm can only find convex or spherical clusters, so for the research in this paper, the CURE algorithm and the DBSCAN algorithm are more suitable. As mentioned above, when collecting RSSI data, it is inevitable to collect noise. Therefore, we hope to find an algorithm that can de-noise the data while dividing the data, and the DBSCAN algorithm is more efficient in dealing with noise than the CURE algorithm. From the shape of the dataset clusters and the efficiency of noise processing, the DBSCAN algorithm is a more suitable algorithm.

The flow of the DBSCAN algorithm is as follows:(1)Scan the entire dataset, find any core point, and expand the core point. The augmentation method is to find all density-connected data points starting from this core point. Traverse all the core points in the neighborhood of the core point and look for points that are densely connected to these data points until there are no data points that can be expanded. The boundary nodes of the final clustered clusters are all non-core data points.(2)Rescan the remaining data set to find the core points that have not been clustered and repeat the above steps to expand the core points.(3)Until there are no new core points in the dataset. Data points in the dataset that are not included in any clusters constitute noise.

The DBSCAN algorithm can divide the data into core points, edge points and noise points according to the density. The difference between core points and edge points is that the data density of the two is different. There is an evaluation score inside the DBSCAN algorithm. After the operation is completed, each data will be scored. Data with a score greater than 0 is divided into core points, data with a score less than 0 is divided into edge points, and data with a score equal to 0 is divided into noise points. We believe that the RSSI data of places often used in life must be dense and high intensity, so according to this feature, we use the DBSCAN algorithm to divide high-density places into core points.

Using the DBSCAN algorithm, the data set can be divided into core points and edge points, and different positioning algorithms are used in different areas. Two different positioning algorithms are applied to the system, which can well solve the balance between positioning time and positioning accuracy. To avoid the problem of long positioning time in less commonly used areas.

### 3.3. The Positioning Process of the RF Algorithm

In the selection of the positioning algorithm for discrete points, it is necessary to choose an algorithm that can meet the needs: not only to meet the needs of relatively high positioning accuracy and short time, but also when the sample dimension is very high, the selected algorithm still has a good positioning effect. After a series of investigations and tests, the RF algorithm is finally used as the edge point positioning algorithm.

The RF algorithm is an ensemble algorithm belonging to the Bagging type. As shown in Figure 2, by combining multiple weak classifiers, the final result is voted or averaged, so that the result of the overall model has high accuracy and generalization performance. It can achieve good results, mainly due to “random” and “forest”; one makes it resistant to overfitting, and one makes it more accurate [59,60].

In the construction process of the RF algorithm, there are two main steps:(1)Assuming that the offline database is the training data set, part of the data is randomly selected, replaced N times, and a decision tree is constructed. This randomness ensures that each decision tree has a different focus on data learning and ensures independence between trees.(2)Assuming that the number of different features of the training data set is D, select some features randomly as E, and ensure that each time E is less than D, and the E feature is the decision condition of the decision tree. The number of feature selections determines the effectiveness of random forests. In other words, if it is too small, the classification accuracy will be low, and conversely, if it is too large, the independence between trees will be reduced. With this randomness, decision trees have good independence and appropriate classification accuracy.

When the online server receives the RSSI value, each decision tree will have a decision result and vote, and the final result is the pattern of the voting results of all decision trees.

### 3.4. DNN Localization Algorithm

If the UJIIndoorLoc dataset is directly used without any processing, it will inevitably introduce redundant information due to too many features. The best way to remove redundant information is to use Principal Component Analysis (PCA) for dimensionality reduction, but PCA works well for linear data, and RSSI values are nonlinear data, so this paper adopts stacked autoencoder (SAE) for dimensionality reduction.

An autoencoder can be thought of as a system that restores its original input. As shown in Figure 3, a simple autoencoder model consists of an encoder and a decoder, with numbers such as 520 representing the number of neurons. The process can be understood that the encoder first transforms the input signal X into the encoded signal Y through functional transformation, and the task of the decoder is to represent the original input of the encoded signal Y in another form. If the input signal is encoded with different X, the system can restore the input signal according to Y, then Y has carried all the information of the original data, but it is output in another form, which is the feature extraction.

A single autoencoder is a three-layer network in the shape of X→h→X′, which can learn a feature change h=f(x) to transform its initial information. However, the output information $X^{\prime }$ is only meaningful for training the auto-encoder, therefore, in practice, a new autoencoder is trained with h as the initial information to obtain a new feature expression, and so on, the output of the previous layer of encoders is used as the input of the latter layer of encoders. This forms the so-called stacked autoencoder and completes the dimensionality reduction process.

Since the localization task is regarded as a classification task, the DNN classifier is used as the localization algorithm and the forward propagation algorithm is adopted. In DNN, the layers are fully connected, that is, like a perceptron, any neuron in the i layer is connected to any neuron in the *i* + 1 layer. As shown in Figure 4, it is a small part of the DNN network.

Assuming that there are m neurons in the *L* − 1 layer, the output $a_{j}^{i}$ of the *j*th neuron in the *L*th layer can be obtained from the Equation (2):(2)$$a_{j}^{l}=\sigma (z_{j}^{l})=\sigma (\sum_{k=1}^{m}w_{j}^{l}{}_{k}a_{k}^{l-1}+b_{j}^{l})$$

In Equation (2), $b_{j}^{l}$ is the bias variable of the *j*th neuron in the *L*th layer, $a_{k}^{l-1}$ is the output corresponding to the *k*th neuron in the *L* − 1 layer and $w_{jk}^{l}$ is the matrix corresponding to the *k*th neuron in the *L* − 1 layer and the *j*th neuron in the *L*th layer, this matrix is the weight between network connections, and by adjusting the weights between neurons, the difference between the actual output vector and the expected output vector of the network can be minimized.

In our system, as shown in Figure 5, HL stands for hidden layer. The pre-trained stacked autoencoder is connected to the classifier to complete the localization task. The figure shows a classifier with two hidden layers, and the numbers, such as 520 and 256 in parentheses are the number of neurons in each layer.

In neural networks, the learning rate is one of the difficult parameters to set. If the learning rate is too small, the parameters with large gradients will have a very slow convergence rate; if the learning rate is too large, the optimized parameters may Instability will occur. Therefore, the neural network in this paper adopts the Adam optimizer and integrates the adaptive learning rate algorithm into the system.

Suppose the initial learning rate is θ, the step length is ε, the first and second moments estimate the decay rate is $\rho_{1}\;\mathrm{and}\;\rho_{2}$, the Minimum constant for numerical stability is δ, and the Updated learning rate is $\theta^{\prime }$. The algorithm steps are as follows:

(1)Initialize first-order and second-order moment variables S=0,R=0, Initialization time t=0.

(2)A small batch containing m samples $\left\{x_{1},....,x_{m}\right\}$ was collected from the training set; corresponding target is $y_{i}$.

(3)The gradient is calculated according to the Equation (3) on the basis of the mini-batch data.


(3)
$$g\leftarrow \frac{1}{m}\Delta \theta \sum_{i}L(f(x_{i};\theta ),y_{i})$$


(4)Refresh the time according to Equation (4).


(4)
t←t+1


The gradient obtained by Equation (3) is substituted into Equations (5) and (6) to update the biased first-order moment estimation and the biased second-order moment estimation.
(5)$$S\leftarrow \rho_{1}S+(1-\rho_{1})g$$
(6)$$R\leftarrow \rho_{2}R+(1-\rho_{2})g⊗g$$

(5)The updated partial first-moment estimation is substituted into Equation (7) to achieve the correction of first-moment error.


(7)
$$\hat{S}\leftarrow \frac{S}{1-\rho_{2}^{t}}$$


(6)The corrected first-order moment error and biased second-order moment estimation are substituted into Equation (8) to achieve the correction of second-order moment error.


(8)
$$\Delta \theta =-\varepsilon \frac{\hat{S}}{\sqrt{\hat{R}+\delta }}$$


(7)Finally, the learning rate is updated through Equation (9).


(9)
$$\theta^{\prime }=\theta +\Delta \theta$$


The reason why Adam is a better solution than RMSProp is that: In Adam, momentum is directly incorporated into the estimation of the first moment of the gradient. The most intuitive way to add momentum to RMSProp is to apply momentum to the scaled gradient. Second, Adam includes bias correction, which corrects the estimates of the first and second moments initialized from the origin. RMSProp also uses second-order moment estimates, however, the correction factor is missing, and the RMSProp second-order moment estimates may be highly biased at the beginning of training. Therefore, the Adam optimizer is generally considered to be robust to the choice of hyperparameters.

### 3.5. Positioning Process

Combined with the algorithms mentioned above, a WiFi indoor positioning system based on area segmentation is proposed.

As shown in Figure 6, the specific steps are as follows:(1)First normalize the data and then use the DBSCAN algorithm to divide the dataset into three regions: core points, edge points and noise points.(2)The data of core points and edge points are set out by using the set-out method, and the data are divided into training set and verification set, which are used for training and verification of the improved DNN model and RF model.(3)The RSSI value received in the online phase is used as the input, and the calculation input is compared with the Euclidean distance of the core point and the edge point. If it belongs to the core point, the improved DNN is used to complete the positioning task. If it belongs to the edge point, the RF algorithm is used to complete the positioning task.

**Figure 6 sensors-22-07920-f006:**
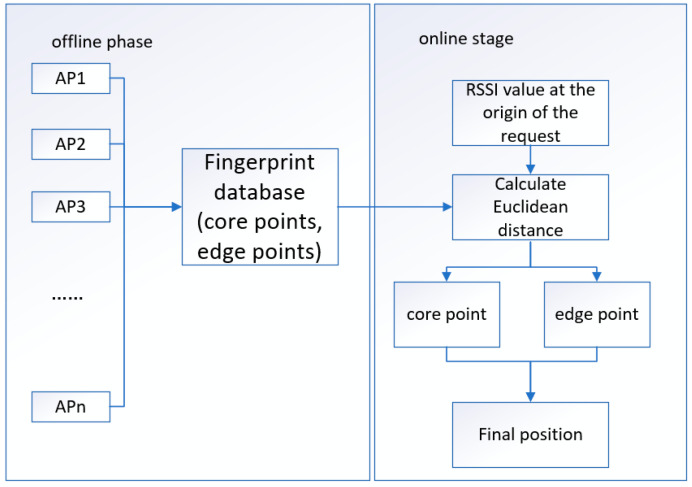
The overall positioning process of the system is divided into offline stage and online stage.

## 4. Experimental Procedure and Results

In this part, the data processing process, edge point positioning process and core point positioning process are introduced and analyzed respectively, and the results are compared with other WiFi-based positioning systems.

### 4.1. Data Processing

In this paper, in order to verify the effectiveness of the system, the UJIIndoorLoc dataset is used as the original data, the program written in Python language is used and simulated on PyCharm2019.

Since the calculation of Euclidean distance is involved in the positioning system, the data is normalized at the beginning. The Z-Score method and the Max-Min method are mainly used for simulation experiments. As shown in Table 1, the accuracy improvement effect brought by the Max-Min normalization method is better than that of the Z-Score method.

In this system, data segmentation is the most important step in the data processing process. The DBSCAN algorithm can not only complete the segmentation of irregularly shaped datasets better than the K-mean algorithm, but there is also an indicator inside the algorithm. When the index is greater than 0, the data points are divided into core points; when the index is less than 0, the data points are divided into edge points, and the rest are noise points. The neighborhood radius and the threshold of the number of data objects in the neighborhood are two important parameters of the DBSCAN algorithm. As shown in Figure 7a, different combinations of the neighborhood radius and the threshold of the number of data objects in the neighborhood can bring different segmentation effects. Since the data density of the UJIIndoorLoc dataset itself is not very large, if the neighborhood radius is too small or the threshold for the number of data objects in the neighborhood is too large, the segmentation effect will be poor, which may lead to data loss. Finally, the neighborhood radius is set to 100~300, and the threshold of the number of neighbor data objects is set to 0~32.

Figure 7b shows the effect of thresholds on the number of data objects in the neighborhood on performance metrics. As can be seen from the figure, the closer the parameter is to 0, the higher the effect index, which means that the more data of the core point, the more segmentation, the better the effect. However, the minimum amount of data in each cluster can not be 1, which will cause each data to be an independent cluster, so we set this parameter to 2. Figure 7c shows the effect of neighborhood radius on the effect index. As can be seen from the figure, there is an improvement process between 200 and 250, and the effect index is the best at 227, so we set this parameter to 227.

As shown in Table 2 above, after the data is processed for normalization and region segmentation, the effect obtained is obvious. For the DNN algorithm, when the positioning accuracy is less than 1.5 m, the probability increases from 72.5% to 98.8%, and When the RF algorithm’s positioning accuracy is less than 1.5 m, the probability increases from 69.6% to 87.2%. We believe that the main reason for the obvious improvement in accuracy is that through region segmentation, the data with the same characteristics are divided together, and the localization algorithm is easier to find similar regions when performing the classification task to complete the localization task.

### 4.2. Edge Point Location Process

For edge points, we expect to find a positioning algorithm that has a short positioning time but is accurate enough to meet daily use requirements and is stable enough. Simulation experiments are performed on net regression, Lasso regression, Bayesian regression, gradient boosting, multilayer perceptron, KNN and RF algorithms.

Lasso regression is a compressed estimation. It obtains a more refined model by constructing a penalty function, so that it compresses some coefficients and sets some coefficients to zero. Therefore, the advantage of subset shrinkage is preserved, which is a biased estimation for dealing with complex collinear data. Elastic network regression is a combination of ridge regression and Lasso regression. It is good at solving models with correlated parameters: lasso regression filters out relevant parameters and reduces other irrelevant parameters; at the same time, ridge regression reduces all relevant parameters. Bayesian regression is a linear regression model solved using Bayesian inference methods in statistics. It treats the parameters of the linear model as random variables and computes its posteriors through priors on the model parameters (weight coefficients). The KNN algorithm is one of the most basic and simple machines learning algorithms, and it can be used for both classification and regression. The eigenvector space is divided by the distance between different eigenvalues of the training data, and the division result is used as the final algorithm model. A multilayer perceptron is a forward-structured artificial neural network that first trains a model, then stores data using weights, and uses algorithms to adjust weights and reduce bias during training, i.e., the error between actual and predicted values. Gradient boosting is an ensemble learning algorithm and machine learning technique for regression and classification problems that produces predictive models in the form of ensembles of weak predictive models. The main idea is to establish the gradient descent direction of the model loss function before each model is established, that is, to generate the model by optimizing the loss function.

First, perform 100 repeated simulation tests on different positioning algorithms, and take the average value. As can be seen from Figure 8, for the edge point data, the user positioning accuracy of RF algorithm, KNN, gradient boosting and multi-layer perceptron are all the same. It has reached more than 80%. However, since the positioning time of the KNN algorithm is proportional to the amount of data, when applied to practice, the positioning time may be too long due to the large amount of data, so the RF algorithm, gradient boosting and Multilayer perceptron for localization time testing.

After 100 repeated simulation tests for three different positioning algorithms, the mean and variance of the positioning time and positioning accuracy of each algorithm are shown in Figure 9 and Table 3 when the positioning error is 1.5 m. The positioning time is the simulation time recorded in python language during simulation. Although the accuracy of the gradient boosting algorithm is slightly higher than that of the RF algorithm, its positioning time is the longest. The positioning time of the RF algorithm is consistent with the expected target, and the positioning accuracy is slightly higher than that of the multi-layer perceptron algorithm. The variance of the RF algorithm is the smallest, which proves that it has sufficient stability, so the RF algorithm is finally used. This algorithm acts as a localization algorithm for edge points.

In the RF algorithm, the n_estimators parameter is the number of classifiers, which is one of the important parameters of the algorithm. If this parameter is too small, it is easy to under-fit, if it is too large, it will cause too much calculation and increase the positioning time, so you need to choose a moderate value. As shown in Figure 10, when the parameter reaches 14, the accuracy is stable between 86% and 87.2%, so the parameter is set to 22, and the accuracy is 87.2%.

### 4.3. Core Point Positioning Process

For the selection of the positioning algorithm of the core point, considering that there are too many database features, the database needs to be reduced in dimension, so the DNN algorithm with stacked autoencoder is selected as the positioning algorithm of the core point.

When choosing the optimizer and classifier, we simulated and tested different combinations between the Adam optimizer and the GradientDescent optimizer, as well as the softmax classifier and the sigmoid classifier. Table 4 shows the corresponding user positioning accuracy of each combination when the positioning error is 1.5 m. The final choice is Adam optimizer and softmax classifier.

The learning rate is one of the most difficult parameters to set in the neural network. If the learning rate is too small, the convergence rate may be slow. If the learning rate is too large, the optimized parameters may be unstable. Therefore, in the DNN algorithm, by adding an adaptive learning rate, it is possible to set a different learning rate for each parameter participating in the training. Although the adaptive learning rate can solve a lot of troubles, the initial learning rate will still have a great impact on accuracy. Figure 11 shows the accuracy of different initial learning rates on the training set and validation set. The accuracy of 0.007 is 99.7% on the training set and 98.8% on the validation set. Figure 12 shows the loss function values corresponding to different training steps. After the training step is 100, the loss function gradually converges. Finally, set the initial learning rate to 0.007 and the epoch to 150.

It has been proved by 100 repeated simulation tests that the DNN algorithm has a good effect on positioning. Without processing the data, the user positioning accuracy is only 72% when the positioning error is 1.5 m. The core point dataset is used as the positioning data of the improved DNN algorithm, the user location positioning accuracy can reach 98.8% when the positioning error is 1.5 m. Table 5 shows the comparison of the accuracy and variance of the improved DNN and KNN, gradient boosting, and multi-layer perceptron algorithms at the core point when the positioning error is 1.5 m. Since the time of the KNN positioning algorithm will vary greatly according to the amount of data, the time has not been measured. Figure 13 is the time comparison of the multi-layer perceptron algorithm, the gradient boosting algorithm, and the improved DNN algorithm. It can be seen that the improved DNN algorithm is better than other algorithms in time, while ensuring the positioning accuracy. It is worth mentioning that the impact of SAE’s dimensionality reduction processing cannot be seen only from the accuracy. If the data is not dimensionally reduced, it will bring dimensional disaster to the DNN network, and eventually lead to an exponential increase in the positioning time. To sum up, it can be seen that, for the core point, the improved DNN algorithm is obviously better than other algorithms.

## 5. Conclusions

With the improvement of material life, WiFi wireless network has changed from private to public, which provides a foundation for the development of WiFi positioning technology. In people’s life, 80% of the time is spent indoors, so it is particularly important to provide a convenient indoor positioning service.

In most positioning systems, only one positioning method may be used, which leads to the phenomenon that the positioning time is too long in uncommon areas, and the positioning accuracy is not required in these areas. In addition, temperature and humidity, the movement of people, and objects can affect WiFi; thereby, affecting the quality of the database, which may eventually lead to a decrease in positioning accuracy. In this regard, this study proposes a localization method for regional segmentation of the dataset. At the core point, the DNN network with SAE integrated with the adaptive learning rate is used to complete the positioning task. SAE can complete the dimensionality reduction processing well, and the DNN network integrated with the adaptive learning rate can provide high-precision positioning. The edge point itself does not need too high accuracy, so the RF algorithm is used to ensure the positioning accuracy and reduce the system positioning time. Experiments show that the experimental results show that our positioning accuracy does not exceed 1.5 m with a probability of less than 87.2% at the edge point, and the time is only 32 ms; the positioning accuracy does not exceed 1.5 m with a probability of less than 98.8% at the core point. In addition, the variance of core points and edge points is generally small, and the system performance is superior, which can meet the daily positioning needs.

At present, the accuracy of the system at the edge point needs to be improved, and other positioning technologies can be adopted at the edge point, or the WiFi positioning technology can be integrated with other positioning technologies. Therefore, in future work, we should also study other low-power or high-precision positioning technologies, such as BLE technology and RFID technology, while researching fusion positioning technology. While ensuring that the system obtains better positioning performance, the positioning cost is also within an acceptable range.

## Figures and Tables

**Figure 1 sensors-22-07920-f001:**
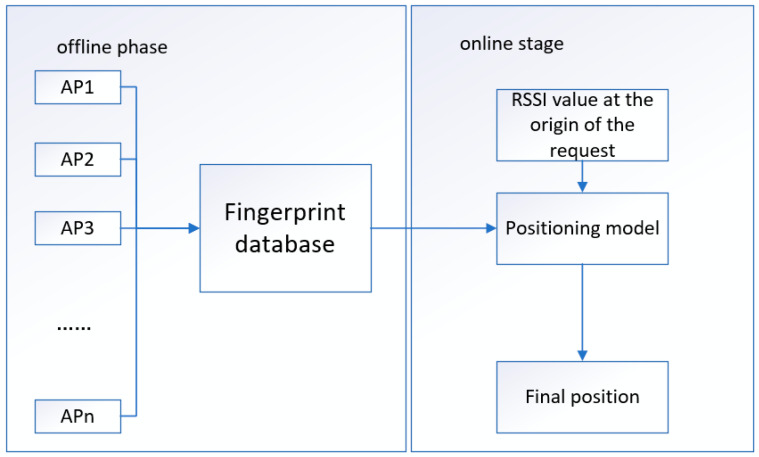
Fingerprint-based WiFi indoor positioning system architecture.

**Figure 2 sensors-22-07920-f002:**
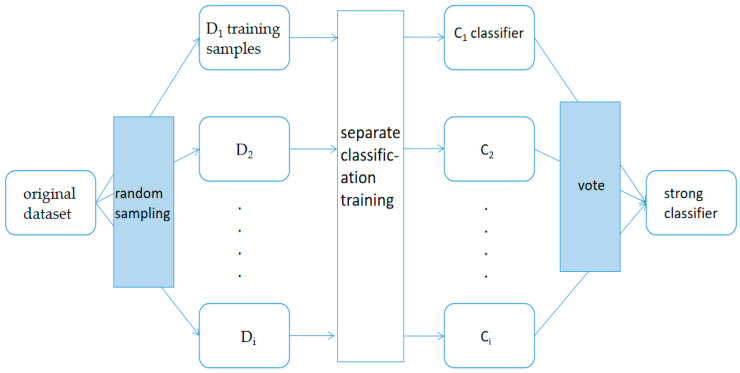
The packet structure of the RF algorithm, by combining multiple weak classifiers into a strong classifier, makes the results have high accuracy.

**Figure 3 sensors-22-07920-f003:**
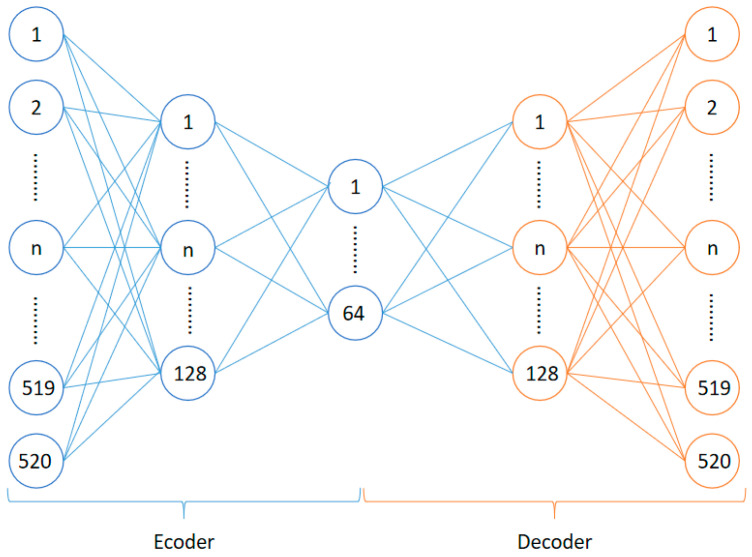
Encoder and Decoder. The output of each layer is used as the input of the next layer, the encoder can achieve dimensionality reduction processing, and the decoder can achieve dimensionality increase processing.

**Figure 4 sensors-22-07920-f004:**
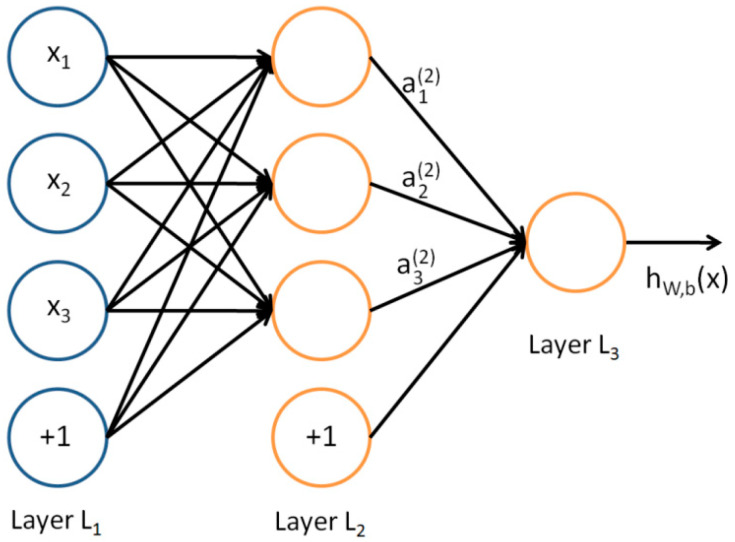
Part of the DNN network.

**Figure 5 sensors-22-07920-f005:**
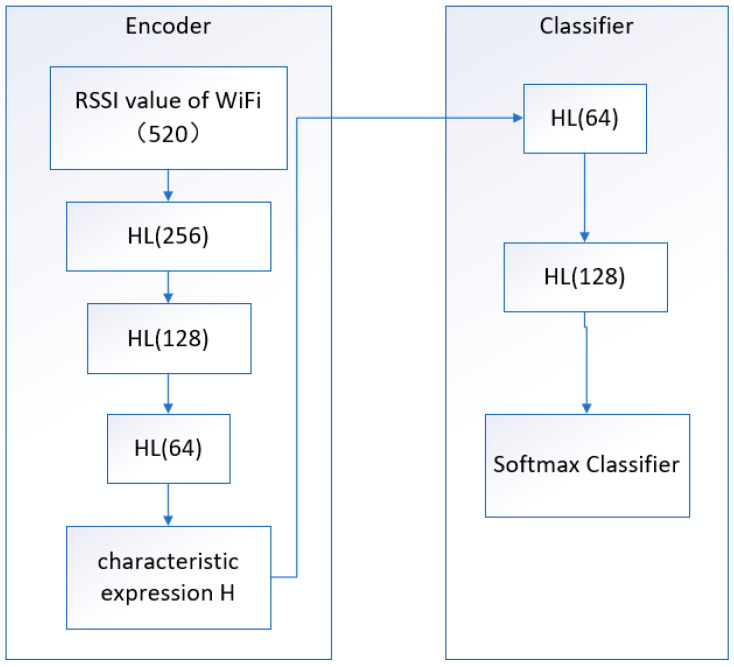
DNN classifier with SAE. SAE is responsible for dimensionality reduction processing, and DNN is responsible for completing the positioning task.

**Figure 7 sensors-22-07920-f007:**
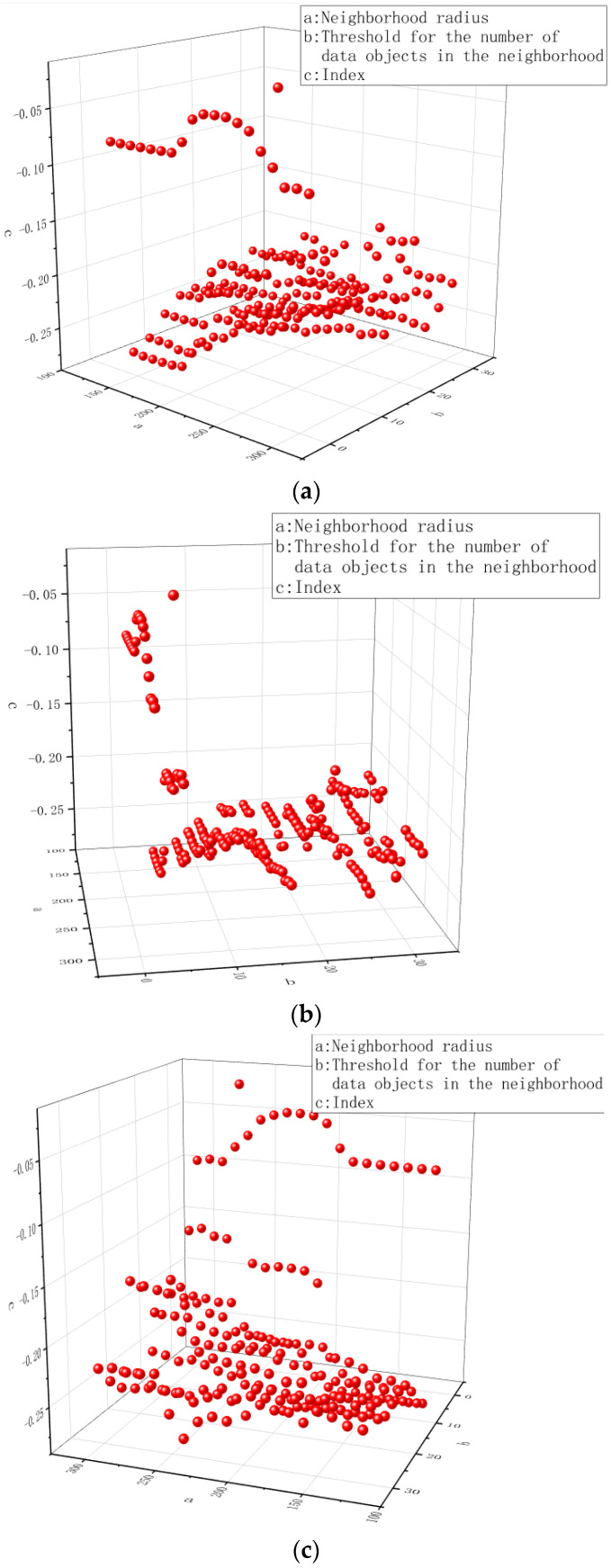
(**a**) The influence of the neighborhood radius and the threshold of the number of data objects in the neighborhood on the indicators of the DBSCAN algorithm. (**b**) shows the effect of thresholds on the number of data objects in the neighborhood on performance metrics. (**c**) shows the effect of neighborhood radius on the effect index.

**Figure 8 sensors-22-07920-f008:**
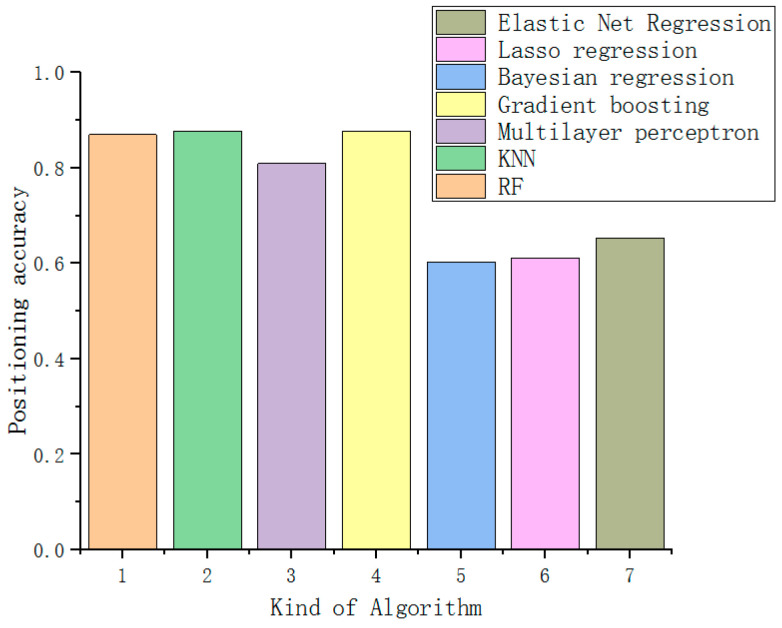
Algorithm accuracy comparison at edge points.

**Figure 9 sensors-22-07920-f009:**
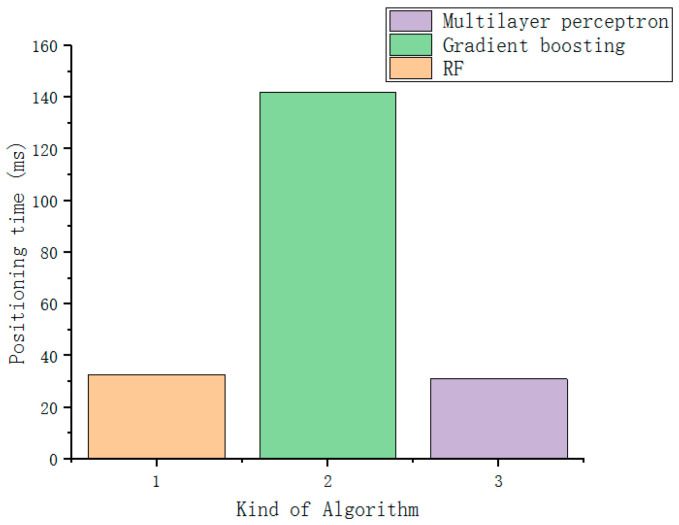
Algorithm positioning time comparison at edge points.

**Figure 10 sensors-22-07920-f010:**
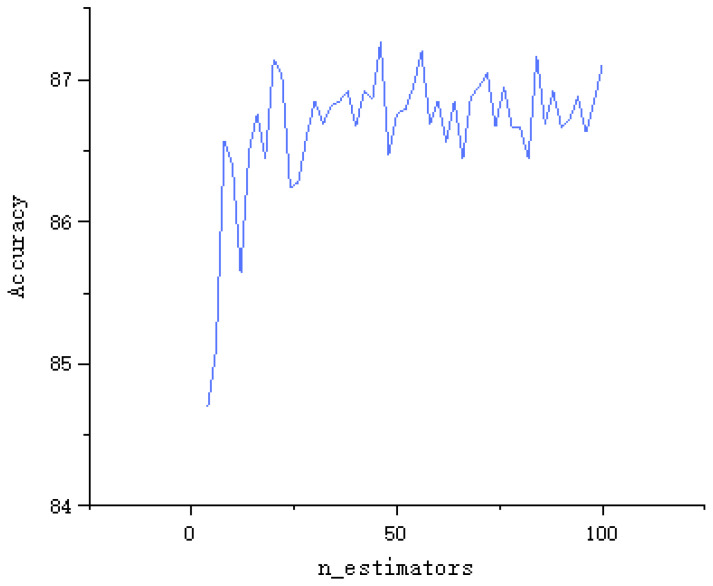
Influence of n_estimators parameter on the accuracy of RF algorithm.

**Figure 11 sensors-22-07920-f011:**
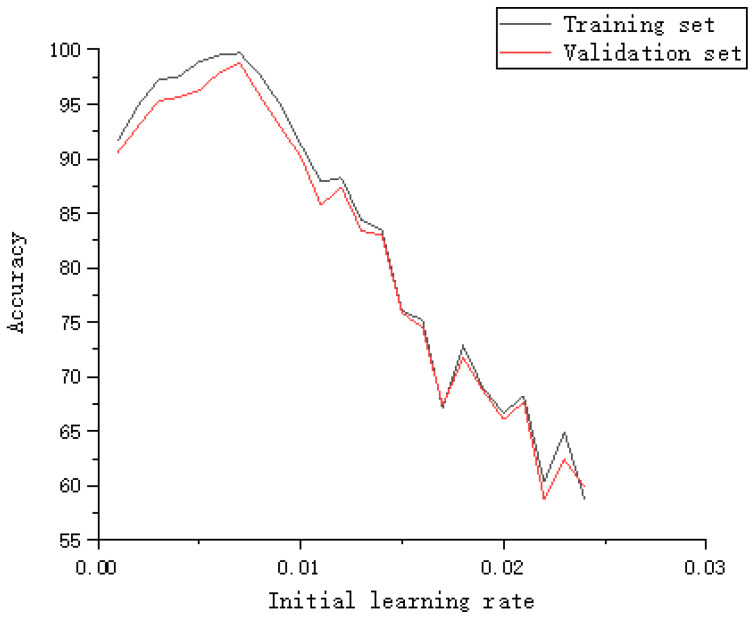
Accuracy corresponding to different learning rates.

**Figure 12 sensors-22-07920-f012:**
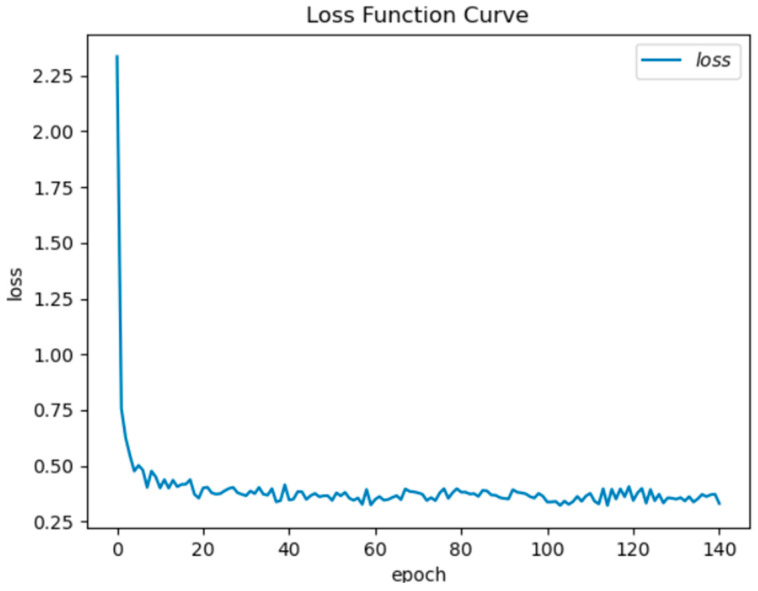
Correspondence map of epoch and loss value.

**Figure 13 sensors-22-07920-f013:**
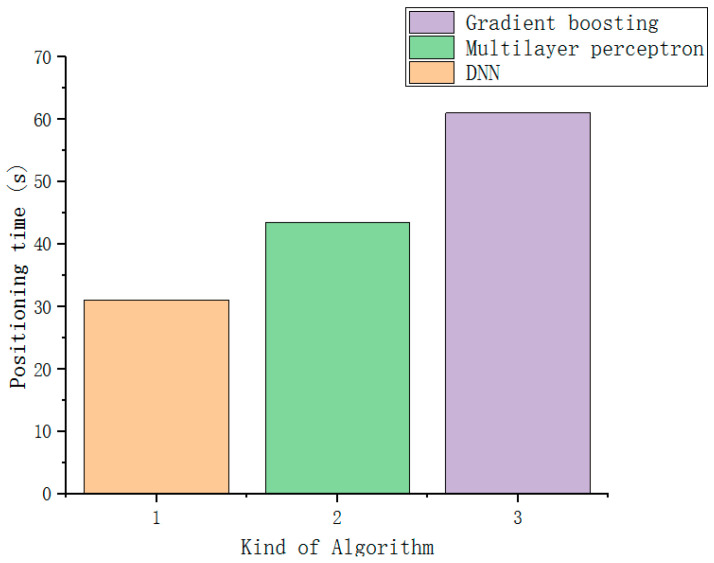
Time comparison of core point algorithms.

**Table 1 sensors-22-07920-t001:** Accuracy corresponding to different normalization methods.

	Core Point	Edge Point
Z-Score	94.9%	82.5%
Max-Min	98.8%	87.2%

**Table 2 sensors-22-07920-t002:** Accuracy before and after data processing.

	Accuracy of DNN	Accuracy of RF
processed data	98.8%	87.2%
unprocessed data	72.5%	69.6%

**Table 3 sensors-22-07920-t003:** Algorithm accuracy comparison.

Kind of Algorithm	Positioning Accuracy	Variance
RF Algorithms	87.2%	0.047
Gradient boosting	87.7%	0.056
multilayer perceptron	81.28%	2.05

**Table 4 sensors-22-07920-t004:** Accuracy corresponding to different combinations.

	Softmax	Sigmoid
Adam	98.8%	58%
GradientDescent	96.5%	53.5%

**Table 5 sensors-22-07920-t005:** Algorithm accuracy at the core point.

Kind of Algorithm	Positioning Accuracy	Variance
Improved DNN	98.8%	0.128
KNN	94%	0.092
Gradient boosting	93.68%	0.055
multilayer perceptron	95.58%	0.084

## Data Availability

Publicly available datasets were analyzed in this study. This data can be found here: http://archive.ics.uci.edu/ml/datasets/UJIIndoorLoc# (accessed on 13 October 2022). The data presented in this study are available on request from the corresponding author. The data are not publicly available due to further research needs.

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
