# Peer review of "WiFi Indoor Location Based on Area Segmentation"

_sensors, 2022, doi:10.3390/s22207920_

Round 1

Reviewer 1 Report

In this paper, the authors apply various data-analysis techniques in order to obtain proper dataset classification and localization by wifi indoor positioning.

The main problem is that this paper lacks clarity in explanation in every item it addresses. Scientific results publication should be clear enough so these results can be reproduced by others. This is not the case in this paper. 

Sections 3 and 4, which are the main contribution of this work, are superficial. There is no clear explanation about why the authors chose a specific classification method, why some of the methods are applied at that particular stage and even the parameters used. Therefore everything looks superficial and a bit amateur. Also, these sections lack a scheme or algorithm that represents the proposed localization system in a clear and precise manner. The results presented, like in figure 7, do not add to the overall clarity and the database set is poorly described. 

In addition, the Introduction and Related Work sections are overinflated in comparison to the methods and results section. The Discussion section lacks a discussion itself. 

I recommend that this paper, as it is now, should be rejected, but can be considered for further publication if the authors perform a great deal of improvement to clarify and structurize their work. 

Reviewer 2 Report

MDPI Sensors Journal (Manuscript ID: sensors-1878231)

Comments to the Author

This paper investigates a Wifi-based indoor localisation system, which uses DBSCAN for region segmentation, a DNN model for user positioning, and the RF algorithm for localisation at the edge points. It is an interesting topic and the paper studies the concept clearly. However, there are several points need to be addressed to improve the quality of the manuscript.

Suggestions to improve the quality of the paper are provided below:

1)     The authors should clearly state the objective of the paper, which is currently missing from the manuscript. Also, the main contributions stated in the Introduction should be rephrased to more clearly highlight its novelty over the existing literature and not just a description of what was done.

2)     I strongly suggest that the authors include a brief description of the different applications of indoor localisation to help to attract other researchers who are working on similar applications to be interested in your paper. For instance, in the building domain, many indoor localisation technologies have been proposed to enable interesting applications such as building energy management, occupancy detection and smart building controls. Please include the following papers to get an idea of these applications:

Indoor localisation for building emergency management

Filippoupolitis, A., Oliff, W., & Loukas, G. (2016, December). Bluetooth low energy based occupancy detection for emergency management. In 2016 15th international conference on ubiquitous computing and communications and 2016 International Symposium on Cyberspace and Security (IUCC-CSS) (pp. 31-38). IEEE.

Indoor localisation for smart plug load control

Tekler ZD, Low R, Yuen C., & Blessing L. Plug-Mate: An IoT-based occupancy-driven plug load management system in smart buildings. Building and Environment, 109472.

Indoor localisation for smart HVAC controls

Balaji, B., Xu, J., Nwokafor, A., Gupta, R. and Agarwal, Y., 2013, November. Sentinel: occupancy based HVAC actuation using existing WiFi infrastructure within commercial buildings. In Proceedings of the 11th ACM Conference on Embedded Networked Sensor Systems (pp. 1-14).

3)     Please provide a more thorough comparison between different indoor positioning technologies to provide a stronger justification on the advantages of using the Wifi technology. For instance, the Bluetooth Low Energy (BLE) technology is viewed as a more energy efficient alternative to the Wifi technology as described in the following papers. Please review and include these below:

Tekler, Z.D., Low, R., Gunay, B., Andersen, R.K. and Blessing, L., 2020. A scalable Bluetooth Low Energy approach to identify occupancy patterns and profiles in office spaces. Building and Environment171, p.106681.

Filippoupolitis, A., Oliff, W. and Loukas, G., 2016, October. Occupancy detection for building emergency management using BLE beacons. In International Symposium on Computer and Information Sciences (pp. 233-240). Springer, Cham.

Other wireless technologies such as RFID has also been adopted by other studies due to its low cost and high localisation accuracy as described in the following papers:

Hahnel, Dirk, et al. "Mapping and localization with RFID technology." IEEE International Conference on Robotics and Automation, 2004. Proceedings. ICRA'04. 2004. Vol. 1. IEEE, 2004.

Li, N. and Becerik-Gerber, B., 2011. Performance-based evaluation of RFID-based indoor location sensing solutions for the built environment. Advanced Engineering Informatics25(3), pp.535-546.

4)     Please provide an explanation on how the DBSCAN algorithm is more superior than the other clustering algorithms (i.e., Kmeans, BIRCH, and CURE), and why it is suitable for this problem. Also, it was mentioned that the neighbourhood radius and threshold for number of data objects are important parameters of the DBSCAN algorithm. However, it is unclear what is the rationale for setting the neighbourhood radius and threshold for number of data objects at 227 and 2, respectively.

5)     Please clarify the difference between edge point and core points, as well as how they are identified.

6)     I suggest that the authors also include an ablation analysis to evaluate the impact of the data processing algorithm and the dimension reduction step.

7)     Some minor feedback to also take note:

·       The data description section should be shifted to the beginning of Section 3.

·       In the related literature section, instead of saying “Reference [X] proposed”, it is more common to directly indicate the authors’ name (i.e., Lee et al., proposed…)

·       Some of the sentences are written in an informal tone and should be corrected to a formal tone.

·       The figure caption can be more descriptive instead of “Bagging Structure” or “Encoder and Decoder”.

·       The axis labels for Figure 7 is very hard to read.

·       Please provide a short description for each baseline algorithms used to perform edge point localisation (i.e., net regression, Lasso regression, Bayesian regression, 364 gradient boosting, multilayer perceptron, KNN).

·       How is positioning time from Figure 9 and 13 calculated?

·       The last section should be named “Conclusion” instead of “Discussion”.

Reviewer 3 Report

This paper proposes a structured system that uses the DBSCAN algorithm to segment regions and realizes regionalized positioning. The Deep Neural Networks (DNN) incorporate an adaptive learning rate which is used for positioning at the core point. The Random Forest (RF) algorithm is used to complete the positioning at the edge points. The experiments verified on the UJIIndoorLoc dataset show good results with higher positioning accuracy and faster positioning time than some positioning algorithms.

The topic is interesting and relevant, but the paper's clarity and completeness have to be improved.

1.     English must be improved to increase the readability of the paper.

2.     The problem statement must be improved to enhance the paper’s clarity. For example, in the second paragraph of the introduction section, “… However, due to the instability of WiFi[18], many factors can interfere with it, so there has…”, what are the factors? In the third paragraph of the introduction section, “… because when collecting data, there are many factors that can affect the RSS value.”, what are the factors?

3.     The author(s) shall briefly describe the methods used in the new proposed system in the introduction section. This work only lists the system methods, e.g., DBSCAN, SAE, and DNN, in the main contributions at the end of the introduction section. However, we don't know how the methods correspond to and solve the problems mentioned in this paper.

4.     The author(s) shall take care of the usage of abbreviation, e.g., what’s the “ RSS ”, “ RSSI ”, “ RSSi ”, “ RSS’ ”?

5.     The author(s) shall explain the meanings of the parameters “520”, “128”, “64” in DNN localization algorithm.

6.     The author(s) shall explain the meanings of the parameter “W” in equation (2). What is the W matrix? What is the purpose to design this matrix?

7.     The author(s) shall write a Conclusion section that concludes the results of this work. The conclusion section also shall describe the limitation and future works. It seems not good that the author(s) made a conclusion and give future works in the discussion section.  

Round 2

Reviewer 2 Report

Thank you for addressing my concerns in this manuscript.

Currently, the required changes have been successfully made by the authors. There is one minor comment regarding my comment (comment 2) in the reference list. I noticed that the reference below has been re-indexed and therefore changed. Please update the previous version of the below reference in the manuscript:

Comment 2

I strongly suggest that the authors include a brief description of the different applications of indoor localisation to help to attract other researchers who are working on similar applications to be interested in your paper. For instance, in the building domain, many indoor localisation technologies have been proposed to enable interesting applications such as building energy management, occupancy detection and smart building controls. Please include the following papers to get an idea of these applications:

Indoor localisation for smart plug load control

Tekler ZD, Low R, Yuen C, Blessing L. Plug-Mate: An IoT-based occupancy-driven plug load management system in smart buildings. Building and Environment. 2022 Aug 17:109472.

Reviewer 3 Report

The revision of this paper is OK. It can be accepted. 
